# Experimental Investigation of Coastal Flooding Hydrodynamics Using a Hybrid Defense System

**Abbas Yeganeh-Bakhtiary** [1,2,*], **Mohammadreza Kolahian** [2] **and Hossein Eyvazoghli** [2]

1   Department of Civil Engineering, University of Kurdistan Hewler (UKH), Erbil, Iraq
2   School of Civil Engineering, Iran University of Science & Technology (IUST), Tehran 13114-16846, Iran;
    mr.kolah@gmail.com (M.K.)
*   Correspondence: abbas.yeganeh@ukh.edu.krd

**Abstract:** Recent studies indicated that coastal green belts could not provide proper protection from extreme coastal flooding. Recent studies recommend employing a compound defense system of natural and artificial structures for extreme hazards. In this study, we introduce a new compound defense system consisting of coastal mangrove trees combined with reef ball modular structures. A series of laboratory experiments were conducted to investigate drag force reduction through the hybrid defense system. The hybrid defense system was subjected to a surge-type flow generated by a quickly lifting gate in a laboratory water tank. Within the experimental framework, the hydrodynamics of coastal flooding were described by the characteristics of the surge bore and the absorbed drag force. The obtained results show that the hybrid system effectively enhanced the absorbed bore drag forces and significantly improved the flow-damping performance.

**Keywords:** coastal flooding mitigation; coastal protection; hybrid defense systems; mangrove tree; reef ball





## 1. Introduction

Recent studies indicate that coastal flooding hazards are expected to increase in the near future and developing coastal resilience measures is a priority for endangered countries [1–5]. When tsunamis and storm surges occur, they are both amplified considerably in shallow water and induce flooding in coastal areas. While coastal flooding due to tsunamis and storm surges can result in similar inundation depths and extents, there are important differences between them. Tsunamis usually consist of several waves and affect much larger areas of coastline. Tsunamis last from 10 min to 2 h, and propagated much faster than storm surges, whereas storm surges typically occur over longer periods, ranging from 1 to 12 h, and have only a single peak. Both events can cause significant damage to coastal assets due to the massive force exerted by the rapidly passing current; however, tsunamis are much more energetic and can exacerbate the impacts of coastal flooding [1,6,7].

One method used for decades to reduce exposure to coastal flooding was to build infrastructure such as seawalls, coastal dikes, and other structures, known as hard defense systems (HDSs). However, HDS construction needs large capital investment, and such hard interventions may even aggravate other pressing coastal problems due to climate change (see, among others, [6–9]). On the other hand, soft defense systems (SDSs) such as coastal mangrove forests not only provide coastal protection but also can provide self-repair for post-flooding damages [10,11].

At the same time, the capacity of coastal forests to mitigate flooding has been widely investigated [12–16]. Due to their ecological parameters (species, age, density, and width), coastal forests induce hydraulic resistance, partially reflect the surge, and provide coastal protection by a reduction in flow velocity and height. Furthermore, mangroves, through their complex root system, may slow down coastal erosion by increasing sediment accretion through enhanced friction and flow modification, which leads to coastline expansion [17,18].

Some observations, however, indicate that coastal forests do not sufficiently protect coastal areas from destruction in extreme flooding events [18–21]. Also, field observations have revealed that low-density mangrove forests cannot fully protect assets and suffer from flooding disturbance; mangroves are uprooted, and lose their ability to mitigate disasters, with the damage increased by the resultant driftwood [22–25]. Therefore, employing compound defense systems (CDSs) are already implemented in many different forms to improve flood protection.

A combination of a coastal forest with embankment systems has already been implemented in many different forms to improve coastal flooding protection [16,26,27]. For example, Tanaka et al. [26] compared coastal forest performance with the single tree defense system and proposed a formula for the drag force and moment reduction in compound defense systems. A system of a coastal forest with a moat for reducing the inundation energy was studied as well [28]. Soon afterwards, a hybrid system comprising a coastal forest with a double embankment system was investigated to improve the prevention of coastal inundation [16]. Later, a new coastal forest combined with an embankment and a moat was introduced to provide a better coastal flooding defense system [29,30].

Detailed observation of recent extreme coastal flooding events indicates that compound defense systems can still be destroyed by substrate erosion, producing large amounts of driftwood caused by the destruction. Moreover, the lack of space for implementing such compound approaches in many urbanized areas could be challenging. Therefore, the applicability of the described CDS methods in all environments is highly uncertain [31]. Using reef balls can provide a solution for the space issue and the stabilization problem of coastal trees to prevent uprooting, and improving low-density mangrove performance in coastal protection.

Reef ball structures have been employed as permeable submerged breakwaters, and their ability in wave dissipation due to hydrodynamic interaction with water waves was investigated [32–35]. Krumholz and Jadot [36] studied planting juvenile mangroves in miniature reef balls to restore them in high-energy environments. Using this technique in a pilot test and collecting field results, the mortality rate of the planted mangroves was successfully decreased. The porous structure of the reef balls increases the absorbed drag force, creates turbulence, and prevents the generation of driftwood due to uprooting by intermingling the tree roots. Moreover, reef balls can play a protective role in the growth of young mangroves and in maintaining them against sea waves and flood surges [36].

Thus, we present a new solution to improve coastal forests' protective role against flooding hazards by combining forest trees with reef ball structures to mitigate the damages caused by coastal flooding. The main objective of this study was to experimentally investigate the absorbed drag forces, inundation velocity reduction, and hazard depth of the hybrid defense system. To do so, a water tank that can produce a surge-type flow was employed to explore the flooding hydrodynamics over the hybrid defense system by changing the flow conditions. The mitigation effects were compared paying special attention to the reflection bore, transmission bore, and drag force and moment reduction.

## 2. Materials and Methods

In this study, we aimed to investigate the effectiveness of the proposed compound defense system, which combines coastal trees with reef balls, in enhancing the resilience of coastal forests against the impact of flooding. The characteristics of flooding were examined when induced surges passed through the experimental CDS model to investigate CDSs' effectiveness in mitigating the drag force, inundation velocity, and depth hazards of tsunamis and storm surges. The experiments and employed instrument configurations, measured parameters, and evaluation criteria are presented in the following sections.

### 2.1. Experimental Apparatus and Procedures

The experiments were conducted in a water tank of 18.0 m length, 0.93 m width, and 1.2 m height for the laboratory experiments with different conditions, as shown in Figure 1.

A quickly lifting gate system was designed and installed to generate a surge-type flow to simulate the surge bore. Using a system of springs, the designed gate (Figure 1a) can be opened by 0.8 m in 0.36 s and generated a surge-type flow towards the test section. To simulate the experimental conditions, the forest test section was placed at 6 m downstream of the gate through a fixed slope of 1:10 with a 1.5 length (Figure 1b).

(a)

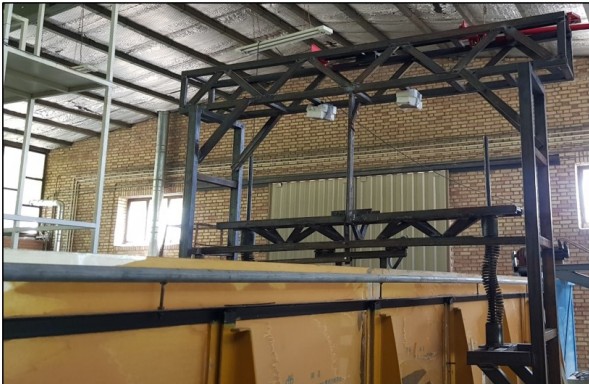

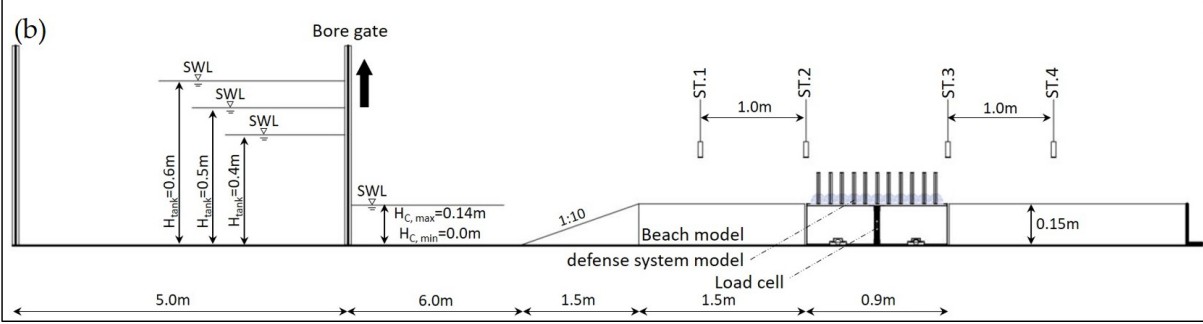

**Figure 1.** Experimental setup: (**a**) picture of designed lifting gate, (**b**) schematic sketch of experiments.

The physical scale of the present study is 1:20 and Froude's similarity law is applied to produce the scaled model; the corresponding water levels and constant water depth were set to 0.40, 0.50, and 0.60 m and 0.0, 0.07, 0.10, and 0.14 m, respectively. The Froude number of generated flows is in the range of 0.73 to 2.36, which offers a suitable range for the experimental modelling of flooding surges [37–41].

Figure 2 shows the arrangement of the coastal trees with the reef ball modules combination. The characteristics of a low-density forest (with a density of less than 0.5%) were used to investigate the proposed solution's performance. The specifications of the mangrove forest of the Makran coast in south-eastern Iran, which consists of Rhizophora species with a density of 1103 units per hectare, were selected [39,40]. Mangrove trees are often replicated using vertical cylinders of Plexiglas cylinders and their flexibility was neglected by various researchers [30,40]. To ensure geometric similarity, Plexiglas cylinders with a diameter of 0.01 m and a height of 0.25 m were placed in a staggered arrangement with a surface density of 0.3%, and they were assumed to be rigid (Figure 2a). Reef balls were made by 3D printing from ABS material with a diameter of 0.1 m and a height of 0.06 m (Figure 2b). As shown in Figure 2, only one forest density, corresponding to a low-density forest, was tested in a 0.9 m width and we focused on the effect of a combination of reef ball modules with single trees to augment their protection role in the examined band.

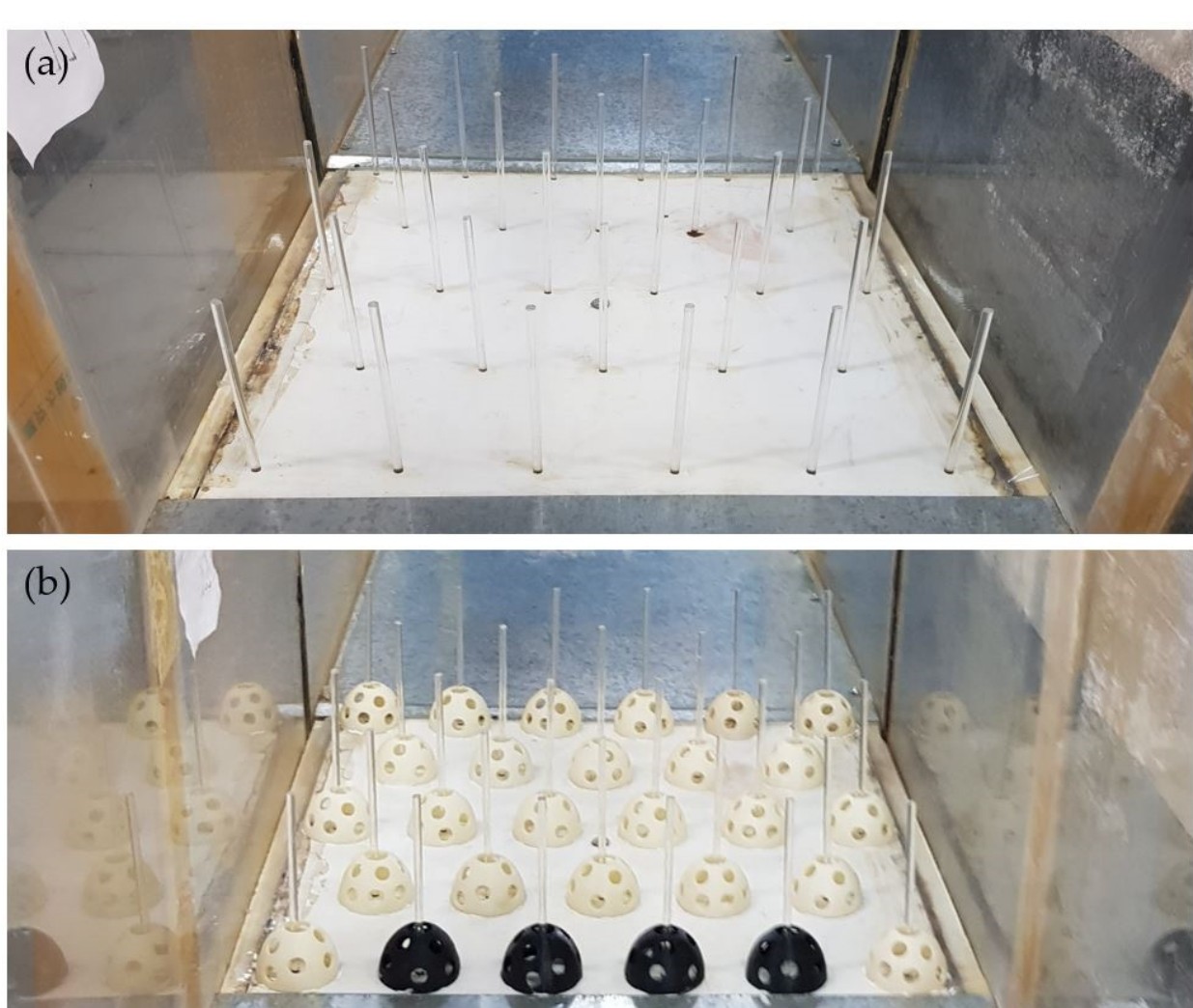

**Figure 2.** Arrangement of combining coastal trees with reef ball modules: (**a**) STDS, (**b**) RCDS.

The experiments were performed in three different conditions: (i) without models (benchmark or control case), (ii) a single tree defence system (STDS), and (iii) a reef ball–tree compound defence system (RCDS). Each case was studied under 12 different flow conditions corresponding to the tank water level ($H_{tank}$ = 0.40, 0.50, and 0.60 m) and a constant water level ($H_c$ = 0, 0.07, 0.10, and 0.14 m). A minimum of three tests were conducted for each case; then their average was presented.

### 2.2. Measurement Method

Four ultrasonic sensors (PEPPERL + FUCHST, UC2000-30GM-IUR2-V15) were placed upstream and downstream of the test section to record the water surface level. To measure the horizontal force induced by the surge flow, a load cell (ZEMIC LOAD CELL, L6D-C3-50 kg) was installed and leaned over the floor of the flume with four bearings installed under the platform to reduce flow friction (see Figure 1b). The force measured in the control case is subtracted from the values measured for the single tree and compound cases to eliminate force due to bed friction.

### 2.3. Examined Parameters

To evaluate experimentally the performance of the proposed compound systems, the following parameters were explored: flow velocity reduction ratio, reflection coefficient, absorbed drag force, drag coefficient, fluid force index, and fluid moment index.

### 2.3.1. Flow Velocity Reduction

The bore velocity reduces as the bore passes from upstream to downstream of the test section. The flow velocity was calculated using the data from ultrasonic sensors and dividing the distance between the two sensors by the time elapsed to pass the wave peak (maximum height of the flow) [42,43]. The flow velocity reduction ratio ($V^*$) was calculated using the following equation:

$$V^* = \frac{V_{Bore} - V_t}{V_{Bore}} \tag{1}$$

where $V_{Bore}$ is the velocity of incident flow based on the measured data of St.1 and St.2 [ms$^{-1}$] and $V_t$ is the velocity of the bore after the platform by considering the measured data of St.3 and St.4 [ms$^{-1}$].

### 2.3.2. Reflection Coefficient

The reflection coefficient ($C_R$) was used to examine the reflected wave characteristics of the studied defense systems as a representative of the reflected energy. We adopted a similar approach to Huang et al. [44] as follows:

$$C_R = \frac{H_R - H_{Bore}}{H_{Bore}} \tag{2}$$

where $H_{Bore}$ is the maximum bore height at St.1 for the test section [m] and $H_R$ is the maximum height of the reflected bore at St.1 [m].

### 2.3.3. Hydrodynamic Force

According to the Morrison equation, the hydrodynamic force exerted on the samples may consist of two components: drag and inertia or impact forces. The spatial inertia ($\partial u / \partial x$) results in the drag force, which is converted to the pressure difference between the two sides of the body, and the temporal inertia ($\partial u / \partial t$) results in the inertia force in Equation (3):

$$F_T = F_D + F_I = \frac{\rho}{2} C_D A_f u^2 + \rho C_M \forall \frac{\partial u}{\partial t} \tag{3}$$

where $F_T$ is total hydrodynamic force, $F_D$ is the drag force [N], $F_I$ is the inertia force [N], $\rho$ is the water density [kg m$^{-3}$], $C_D$ is the drag coefficient, $A_f$ is the frontal area of the models [m$^2$], u is the flow velocity [m s$^{-1}$], $C_M$ is the inertia coefficient, $\forall$ is the volume of the submerged model [m$^3$], and $\frac{\partial u}{\partial t}$ is the horizontal flow acceleration [ms$^{-2}$]. As Imai and Matsutomi pointed out, at the early stage of inundated flow, the inertia force reaches 50% of the maximum drag force; whereas after the bore is developed to the quasi-steady state condition, the drag force becomes dominant. Therefore, the drag force is considered the main component of the total instantaneous hydrodynamic force absorbed by the vegetation model in the test section [45]:

$$F_T \approx F_D = \frac{\rho}{2} C_D A_f u^2 \tag{4}$$

$$C_D = 2F_D / \rho A_f u^2 \tag{5}$$

### 2.3.4. Force and Moment Indices

The effectiveness of the hybrid defense structure was assessed by calculating the drag force and moment reduction. We employed Tanaka et al.'s proposed method and compared compound defense system performance with the single tree defense system [26]. For this purpose, the flow velocity was recorded downstream of the platform ($V_t$) and the water depth at St.4 ($h_t$). The maximum reduction rates of the drag force index (RFI) and the moment index (RMI) are calculated as follows:

$$RFI\% = \frac{\left(V_t^2 \times h_t\right)_c - \left(V_t^2 \times h_t\right)_{max}}{\left(V_t^2 \times h_t\right)_c} \times 100 \tag{6}$$

$$RMI\% = \frac{\left(V_t^2 \times h_t^2\right)_c - \left(V_t^2 \times h_t^2\right)_{max}}{\left(V_t^2 \times h_t^2\right)_c} \times 100 \tag{7}$$

where $(V_t^2 \times h_t)$ and $(V_t^2 \times h_t^2)$ are, respectively, the fluid force and moment indexes; moreover, the subscripts '$c$' and '$max$' represent the value without a control case, and the maximum value in each model case. The higher the *RFI* and *RMI* value, the higher the effect of the hybrid system versus the single tree system [29].

## 3. Results and Discussion

### 3.1. Generated Bore Characteristics

Figure 3 depicts the snapshots of the inundation pattern inside the combined forest with reef balls. As seen, a bore-type flow with a turbulent front was generated in both cases, and the bore height, velocity, and turbulence intensity at the bore front was changing rapidly. The generated bore rushed onto the sloping part of the test section measured at St.1, while the maximum height measured of $H_{Bore}$ is in the range of 0.093 to 0.172 m. Due to the presence of the defense system models, the bore front collision causes the water to splash over the models, accompanied by turbulence in the initial moments.

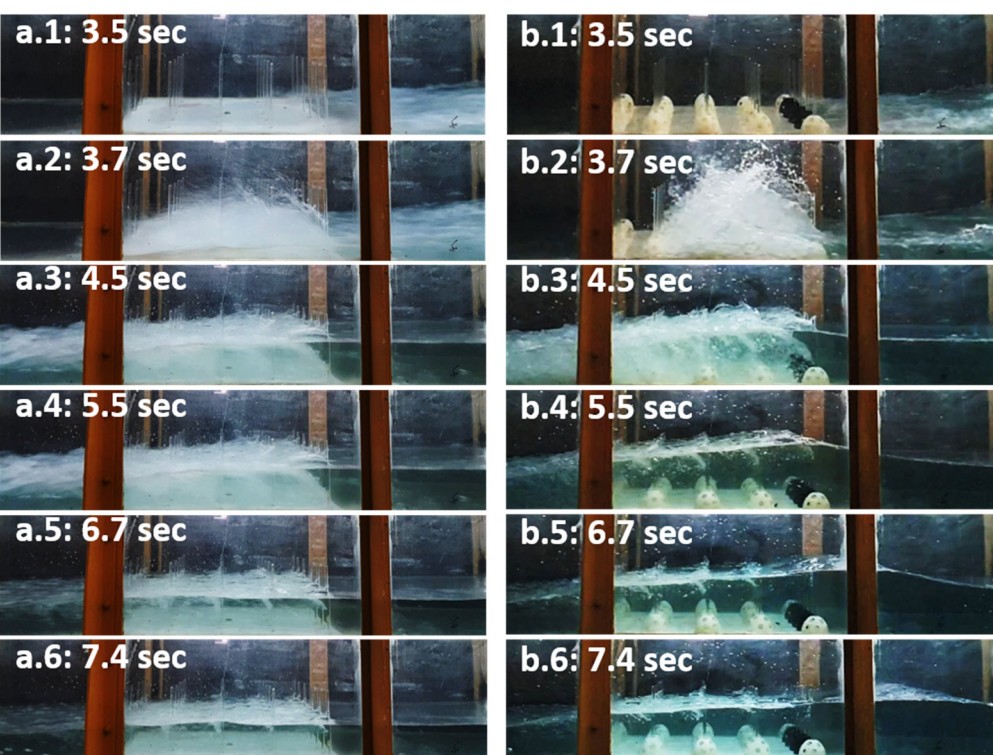

**Figure 3.** Snapshot of inundation patterns: (**a**) STDS, (**b**) RCDS.

In the RCDS case, the splash height is up to two times higher than in the STDS case (see Figure 3(a.2,b.2)). Then, passing the bore through the test section, the perturbations are presented after the samples: in the RCDS model, the perturbations disappeared faster than the STDS model (see Figure 3(a.4,b.4)). In the RCDS model, at 5.5 s, the height of the flow at the beginning of the test section reaches the height of the vegetation tree. Afterwards, a blocked wave returns seaward, which was not observed in the STDS model (Figure 3(a.4–a.6,b.4–b.6)).

Figure 4 depicts the transmitted flow height ($H_t$) versus the incident bore height ($H_{Bore}$). As seen, due to the amplification of the bore height and the streamlines through the test section, the transmitted flow height $H_t$ measured at St.4 does not show a remarkable decrease for the control case compared with both the STDS and RCDS cases. The same

observation is also reported by Zaha et al. [29] and Ahmed and Ghumman [46] for their hybrid defense systems. This might be attributed to the experimental limitations and narrow width of the modeled forest.

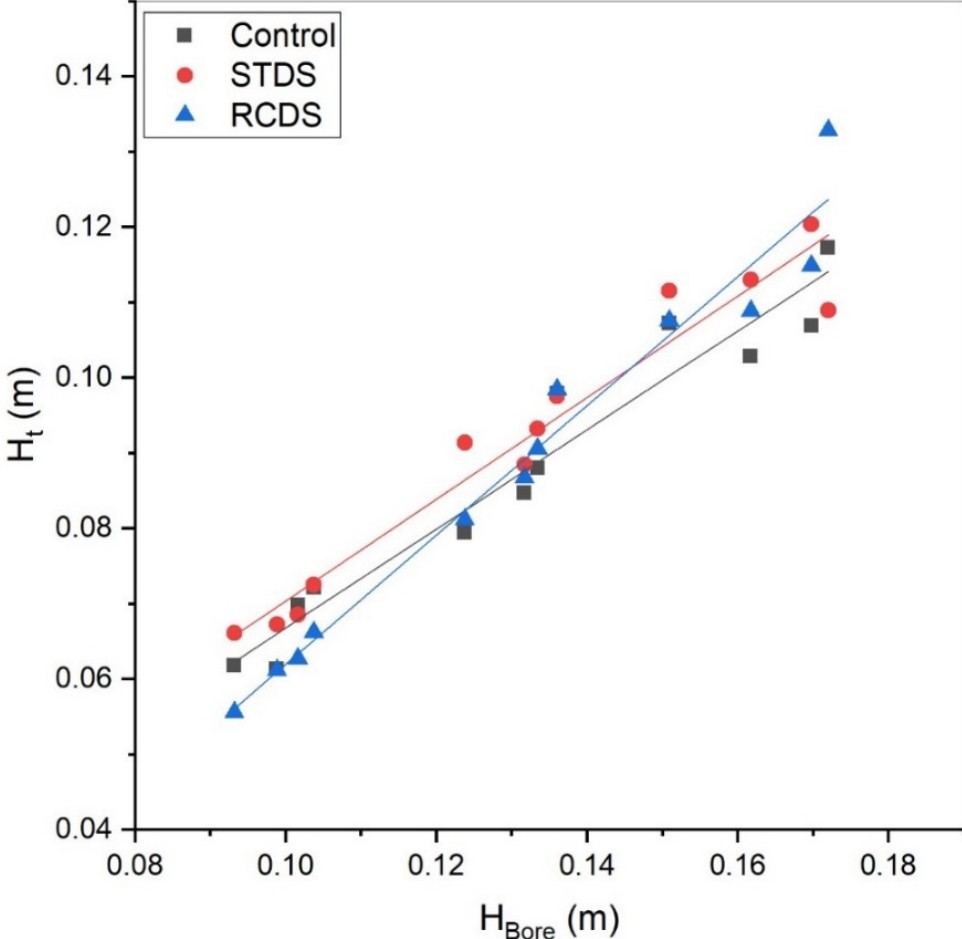

**Figure 4.** Transmitted flow height ($H_t$) versus the incident bore height ($H_{Bore}$).

Figure 5 shows the transmitted flow velocity ($V_t$) versus the incident bore velocity ($V_{Bore}$). As seen, contrary to the bore velocity upstream of the test section, the transmitted velocity decreases downstream of the test section. Flow velocity in the presence of the STDS model and the RCDS model, respectively, decreased by an average of 29.5% and 56.2%. The fitted graphs' growth rate compared to the control case decreased by 47.5% and 68.7%, respectively. A close look at Figures 3–5 indicates that in the hybrid defense system, by changing flow conditions, the drag forces reduced and the reef ball arrangement may influence the mitigation effect of the defense system.

Figure 6 shows the Froude number of transmitted bores versus the Froude number of incident bores. The Froude numbers are estimated as:

$$Fr_{Bore} = \frac{V_{Bore}}{\sqrt{gH_{Bore}}} \tag{8}$$

$$Fr_t = \frac{V_t}{\sqrt{gH_t}} \tag{9}$$

The flow Froude number, calculated using the values of the bore velocity before and after the models ($V_{Bore}$ and $V_t$) and the height values at St.1 and St.4 ($H_{Bore}$ and $H_t$), showed similar changes in the flow Froude number and flow velocity. As seen in Figure 6, the Froude numbers upstream and downstream of the platform were reduced, respectively,

by 49% and 79% for the STDS model and the RCDS model. Also, the growth rate of the fitted graphs decreased by 68% and 91% compared to the control case for the STDS and RCDS models.

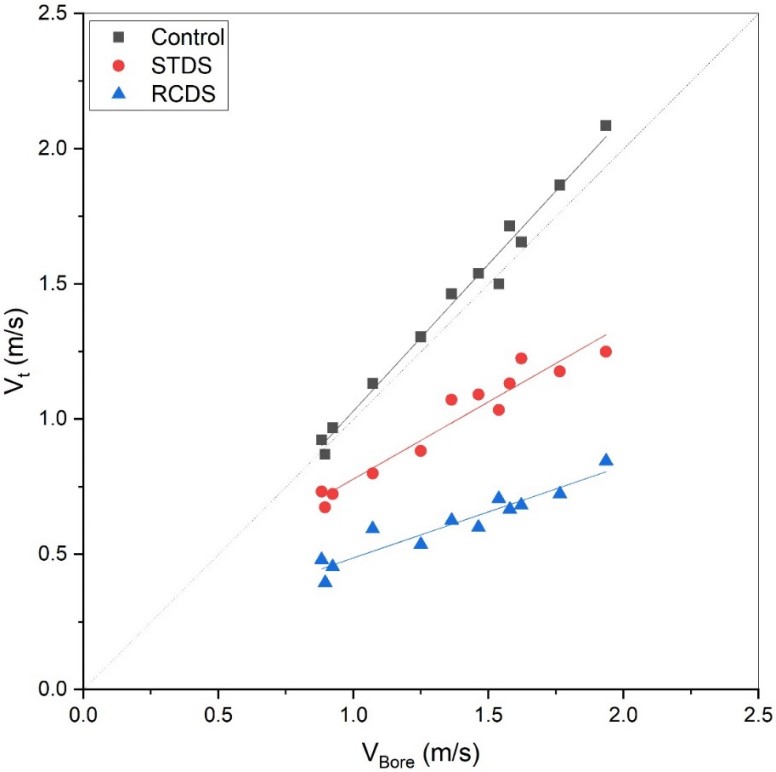

**Figure 5.** Transmitted flow velocity ($V_t$) versus the incident flow velocity ($V_{Bore}$).

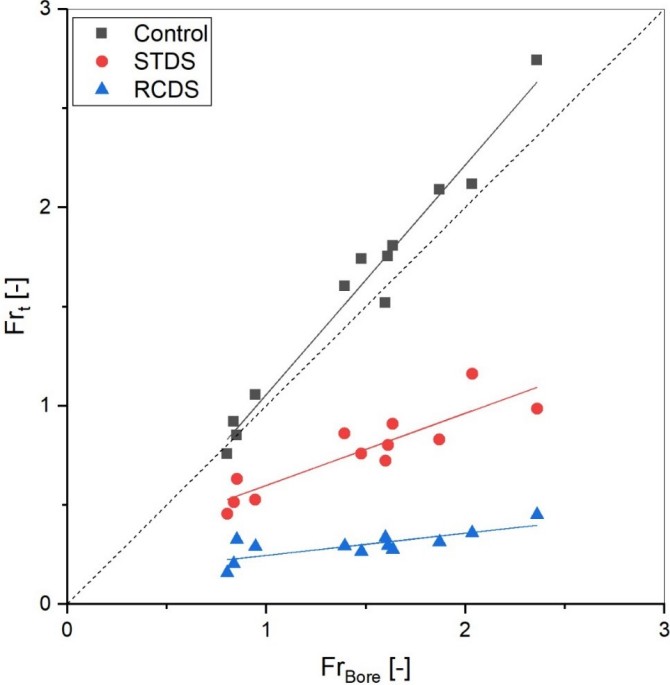

**Figure 6.** Comparison of transmitted flow Froude number ($Fr_t$) with bore Froude number ($Fr_{bore}$).

### 3.2. Drag Forces

The absorbed drag force ($F_D$) of the STDS and RCDS models are depicted in Figure 7. The measured reduced drag force is equal to the maximum shear force absorbed by the defense section when the bore is passing through it. As expected, the drag force is decreased for the tests with defense system models under similar flow conditions. The exerted force imposed on the defense system models is evaluated for the vegetation and reef balls with a similar method to Husrin [47] and Fathi-Moghadam et al. [43]. As can be seen from Figure 7, by increasing the bore Froude number ($Fr_{Bore}$), more force is absorbed by both defense systems.

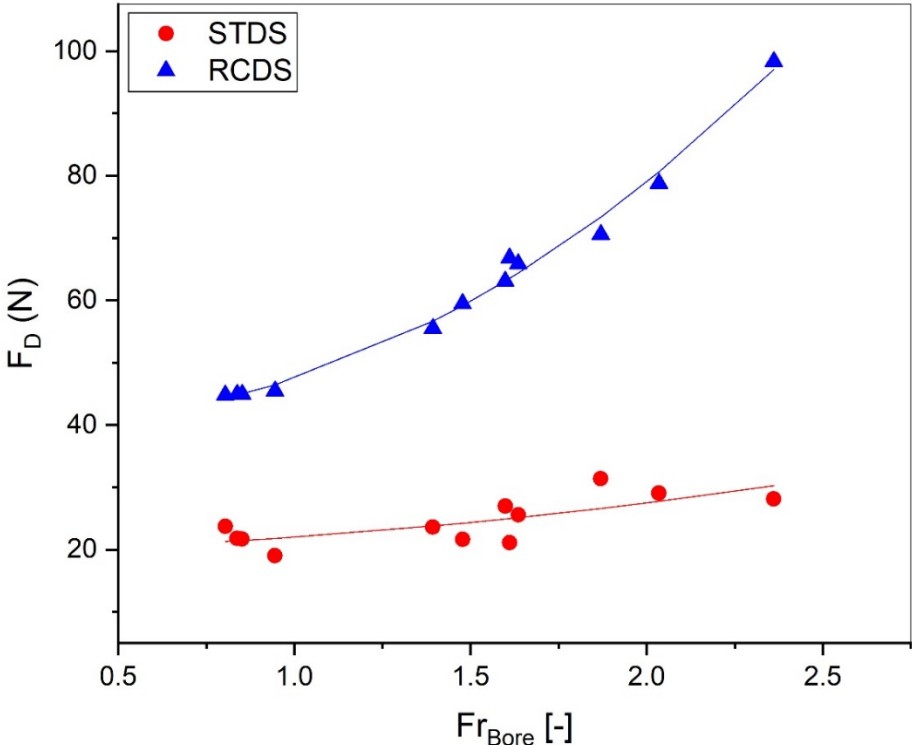

**Figure 7.** The absorbed drag force ($F_D$) versus Froude number.

Furthermore, as the bore height increases, the $A_f$ grows, and more force will be induced into the models. Using the RCDS instead of the STDS in a constant flow increases the absorption of the induced forces. This is partly due to the increased $A_f$, which the reef ball models added to the STDS. The absorbed force by the compound model increased by 89% to 249% compared to the STDS model for the flow with the minimum and maximum $Fr_{Bore}$, respectively. On average, the RCDS produced a 150 percent improvement in flow-damping performance.

The variation of $C_D$ with the incident flow Froude number ($Fr_{Bore}$) is shown in Figure 8 for both defense systems. The drag coefficient ($C_D$) for each defense system is calculated using Equation (5), considering the models' frontal area ($A_f$), incident flow velocity ($V_{bore}$) passed through St.1 and St.2 from ultrasonic sensors and the drag force. As mentioned earlier, the incident bore with a higher Froude number has higher velocities and induces higher hydrodynamics force.

However, the obtained experimental results indicated that the drag coefficient values decrease with increasing Froude number for both defense systems. This decreasing trend was caused by the non-uniform vertical velocity distribution for $Fr > 0.76$ [47]. This leads to a distortion of the velocity in flows with higher Froude numbers that may cause a smaller drag coefficient. The same decreasing correlation has been presented in the equations for drag coefficient and Froude number in previous studies [47–50]. Figure 8 also reveals that the drag coefficient values decrease when the RCDS is employed.

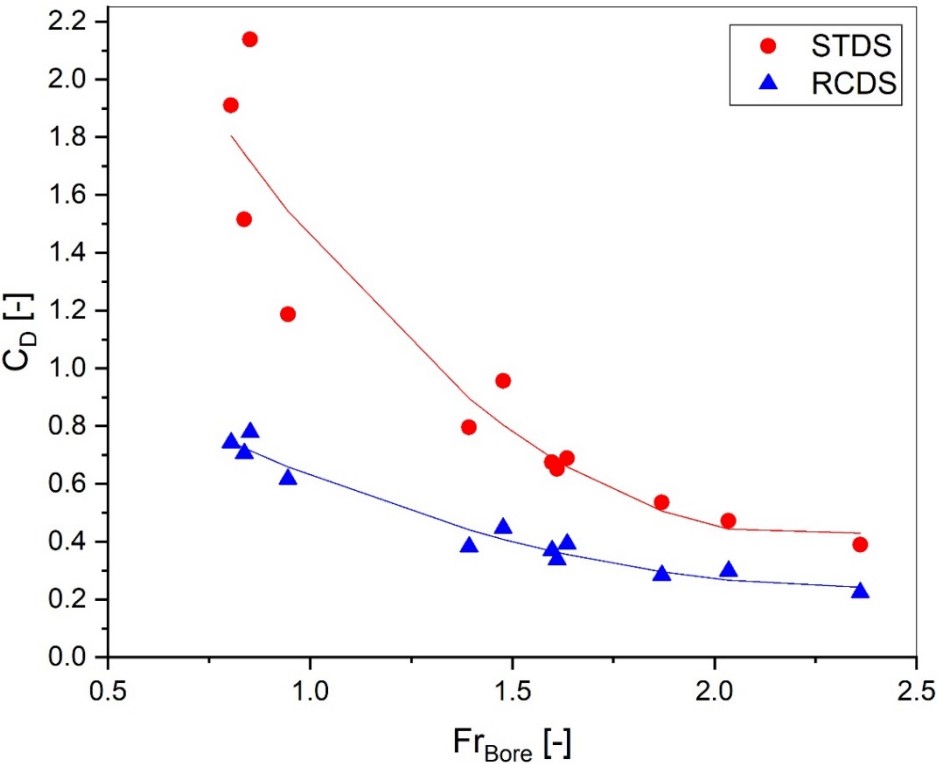

**Figure 8.** The drag coefficient (CD).

The presence of reef ball structures along with trees in the RCDS increases significantly the $A_f$ and covered surface density which causes a decrease in the drag coefficient for the RCDS. This reduction in $C_D$ values is the result of the mutual interaction of the defense system elements. The used equation for the calculation of $C_D$ (see Equation (5)) does not consider the possible interaction of models and their different behavior for different interactions. This reduction in drag coefficient range due to the amplification of vegetation density because of mutual interaction has been reported in previous studies see among others [43,44,46,51].

### 3.3. Defense System Performance

#### 3.3.1. Effect on Velocity Reduction Rate

Bore velocity is one of the determining factors in the kinetic energy of the propagating tsunami and storm surge bore. It is essential to study changes in the velocity in the face of the defense systems' performance. The velocity reduction rate is calculated using Equation (1) and compared for the STDS model and the RCDS model in Figure 9. The values of $V^*$ varied between 17.1% to 35.4% and between 47.6% to 59.0% reduction for the STDS and RCDS, respectively. Using the RCDS, a 111% increase in the defense system efficiency in reducing the bore velocity has been observed, resulting from the flow disruption caused by the increase in $A_f$ and the increase in turbulence behind the test section due to the presence of the porous structures.

#### 3.3.2. Effect on Reflection Coefficient

As discussed in Section 3.1, a reflected wave is propagated seaward in the RCDS model tests, which is not observed in the STDS model. The presence of the compound models causes the formation of this reflected wave. However, the increase in the frontal area results in a water impoundment reaching the top of the mangrove trunks (25 cm), which leads to the reflected wave. The appearance of this reflected wave indicates the improvement of the efficiency of the STDS by combining it with reef ball models.

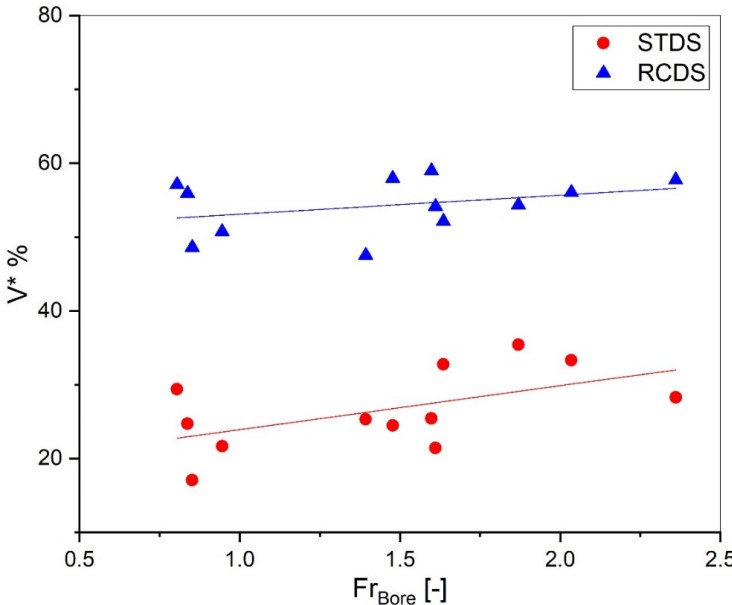

**Figure 9.** The velocity reduction rate ($V^*$).

The relationship between the bore height and the reef ball structure height ($H_{reef}$) is assumed to greatly impact the bore reflection seaward, *RFI*, and *RMI* [29,30]. Therefore, 'the non-dimensional bore height' ($H^*$) was defined by dividing the maximum bore height by the root of the reef ball frontal area as follows:

$$H^* = \frac{H_{Bore}}{H_{reef}} \tag{10}$$

The values of $C_r$ versus the non-dimensional bore height are presented in Figure 10. As seen for the RCDS model, this varied from 0.71 for the lowest $H^*$ to 0.35 for the highest $H^*$. As the height of the incident bore increases, the inundation depth increases, while the frontal area engages a smaller percentage of the current. Hence, a reduction in the value of $C_r$ seems reasonable.

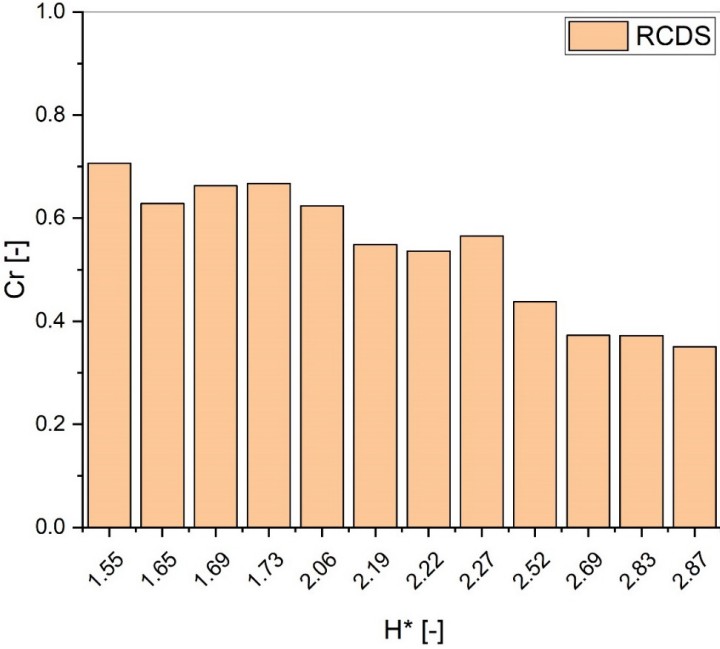

**Figure 10.** The reflection coefficient ($C_r$) versus non-dimensional bore height ($H^*$).

### 3.3.3. Effect on RFI and RMI

The calculated values for *RFI* and *RMI*, using Equations (6) and (7), are shown in Figure 11. By changing the protection system from the STDS to the RCDS, on average, a 70% and 82% growth was obtained for the *RFI* and *RMI* values, respectively, caused by the presence of a reef ball structure collision effect as a result of the streamlining distortion. A similar trend for *RFI* and *RMI* was observed, although the fluid force index ($V_t^2 h_t$) was more affected by the decreased flow velocity after the models than the reduced water depth. The moment index ($V_t^2 h_t^2$) was influenced by the flow velocity and depth in the same order. As mentioned before, the measured values for the flow height after both models did not show a significant difference; therefore, a similar trend of *RFI* and *RMI* seems reasonable.

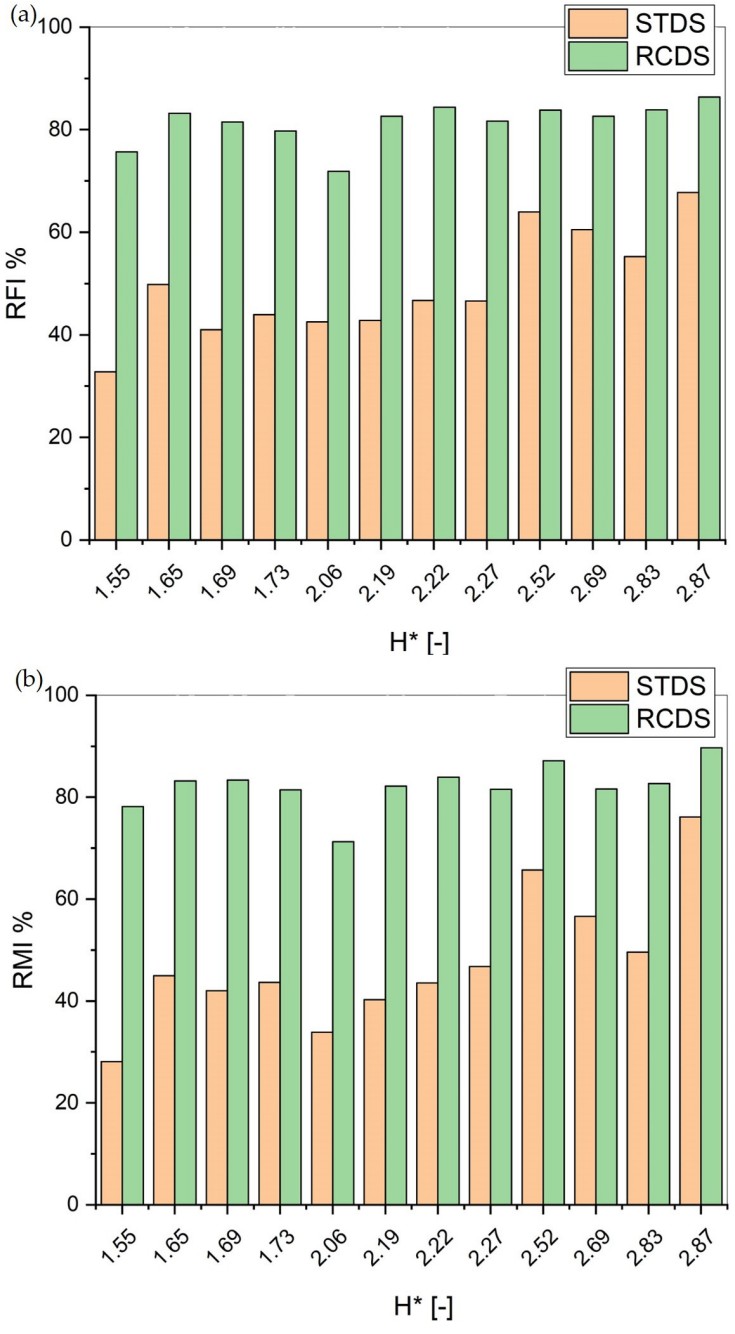

**Figure 11.** Reduction in (**a**) force index and (**b**) moment index.

Zaha et al. [29] introduced the evaluation indices *ERFI* and *ERMI* to compare different arrangements of compound defense systems using the *RMI* and *RFI* obtained for each system under different flow characteristics:

$$ERFI = \frac{\sum_{i=1}^{n}(H^* \times RFI)_i}{\sum_{i=1}^{n}(H^*)_i} \tag{11}$$

$$ERMI = \frac{\sum_{i=1}^{n}(H^* \times RMI)_i}{\sum_{i=1}^{n}(H^*)_i} \tag{12}$$

where *ERFI* and *ERMI* are the evaluation indices for *RFI* and *RMI*, respectively, n is the number of conducted tests (12 tests), and *i* is the indicator of the tests (from 1 to 12). The calculated values for *ERFI* and *ERMI* of the presented RCDS and STDS in comparison with the proposed CDSs of Zaha et al. [29] and Kimiwada et al. [30] for the best order of the combination of vegetation (*V*), embankment (*E*), and moat (*M*) in their studies, are presented in Table 1. $V_{40}$ represents 40 rows of tree models, and $V_{S40}$ represents 40 rows of submerged tree models.

**Table 1.** Comparison of ERFI and ERMI.

| Case | ERFI | ERMI |
|:---:|:---:|:---:|
| STDS | 51.2 | 49.8 |
| RCDS | 81.8 | 82.55 |
| $V_{40}ME$ [29] | 74.3 | 92.4 |
| $EMV_{40}$ [29] | 83.1 | 90.2 |
| $EMV_{S40}$ [30] | 68.7 | 80.8 |
| $EV_{S40}M$ [30] | 66.7 | 74.6 |

As seen, the evaluation indices (*ERFI* and *ERMI*) for CDSs have higher values in comparison with the STDS, and the RCDS model showed 60% and 65% more effectiveness for *ERFI* and *ERMI* compared to the STDS model, respectively. Compared with the previous combination methods presented by Zaha et al. [29] and Kimiwada et al. [30], the presented RCDS model has an acceptable performance in the same order as the compound systems offered by Zaha et al. [29] and produces a better performance than the proposed combination of Kimiwada et al. [30]. Meanwhile, the lower construction cost, the absence of a need to vacant land for construction, and the potential of increasing the stability of the trees make the RCDS a preferable compound defense system over a combination of vegetation with an embankment and moat.

To summarize, the implementation of the RCDS has demonstrated promising results in improving velocity reduction efficiency and reducing absorbed drag force, while potentially reflecting a portion of the incident bore energy. However, the complex shape of mangrove roots presents many construction challenges. The potential solution to tackling these challenges is to plant mangroves in reef balls or place the reef balls around existing young/small mangroves on the coastal site. This approach provides a flexible means of accommodating the natural shapes of the legs and roots, while promoting the mangroves' expansion and resilience. The proposed design contributes to reducing coastal hazards; however, further research is very necessary to look for both an excellent accommodation method to enhance the mangrove resilience and the expansion ranges of the forest width. Proper planning and execution are also critical for ensuring that the mangroves and reef balls are integrated effectively into coastal management strategies to mitigate flood risk.

## 4. Conclusions

The hybrid defense system as a combination of mangrove trees with reef ball modules was introduced and experimentally investigated to increase the coastal flooding mitigation

role of coastal green belts and damage reduction. To attain this objective and considering the experimental limitations, the present study was conducted on a low-density, emerged forest with a width of 0.9 m, comparing the conditions of the forest with and without the combination with reef ball modules. The following conclusions are drawn:

Combing reef ball modules with mangrove trees significantly increases the efficiency of the green belt in velocity reduction and drag force absorption and resulted in average growth of 70% and 82% for the *RFI* and *RMI* indices, respectively.

The results do not show a remarkable decrease in inundation flow depth, which was also reported by previous studies. The obtained results showed that there was not a significant difference in flow height between the cases, which might be attributed to the experimental setup and arrangement of the reef balls and cylinder models. This might be attributed to the experimental limitations and narrow width of the modeled forest.

By changing the defense system from the STDS to the RCDS and increasing the frontal area, a reflected wave can be observed that propagates seaward on the bore surface. The reflection coefficient increases, and a large portion of the incident bore energy was dissipated and reflected upstream.

As result of the inundation velocity reduction and absorbed drag force enhancement, the RCDS can successfully reduce the impact hazard of flooding and reduce hydraulic force, for both tsunamis and fast-moving storm surge events. Moreover, considering the fact that there is no need for vacant space to employ the proposed protection system, the RCDS can provide a compelling solution in urbanized areas.

As result of the inundation velocity reduction and absorbed drag force enhancement, despite no significant difference in flow height, the RCDS can partially reduce the impact hazard of flooding and reduce hydraulic force. Moreover, since there is no need for vacant space to employ the proposed protection system, the RCDS may provide an appealing solution to employ in urbanized areas. The proposed design may contribute to reducing coastal hazards; however, additional research is necessary to validate its efficacy through more extensive measurements with varying degrees of forest coverage.

**Author Contributions:** Conceptualization, A.Y.-B. and M.K.; methodology, A.Y.-B., M.K. and H.E.; validation, A.Y.-B. and M.K.; formal analysis, M.K.; investigation, A.Y.-B. and M.K.; writing—original draft preparation, M.K.; writing—review and editing, A.Y.-B. and H.E.; visualization, M.K.; supervision, A.Y.-B. and H.E.; project administration, A.Y.-B. All authors have read and agreed to the published version of the manuscript.

**Funding:** This research received no external funding.

**Data Availability Statement:** The data that support the findings of this study are available upon reasonable request.

**Acknowledgments:** This study was partially supported by the Deputy of Research and Innovation Management, IUST, Iran.

**Conflicts of Interest:** The authors declare no conflict of interest.

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
