# Peer review of "Experimental Investigation of Coastal Flooding Hydrodynamics Using a Hybrid Defense System"

_water, doi:10.3390/w15142632_

Round 1

Reviewer 1 Report

This is a manuscript presenting an experimental analysis of the performance of a hybrid (mangrove-reef balls) defense system again coastal flooding. The manuscript addresses a topic directly related to hydraulic (coastal) engineering which is one of the scientific targets of Water. In this sense, the manuscript can be of interest for Water readers. In what follow some comments/remarks are given.

 [1] The manuscript needs an in-depth review of the grammar. This will help the authors to better transmit their results to readers. In addition to this, several (many) misprints (e.g. Tanka et al. instead of Tanaka et al.), uncompleted sentences (e.g. To compare the performance of the different compound defense systems two indices of ERFI and EMRI for drag force and moment reduction.), wrong selection of legends (Figure 1. Schematic sketch of experimental setup, (a) picture of designed quickly lifting gate. There is no indication of a and b in the figure). This needs to be fixed. So, please check carefully the text.  

[2] There are some references on the topic that deserve to be included and, more importantly, considered to properly frame the presented work in the context of existing knowledge. Just as an example:

Chang, C. W., & Mori, N. (2021). Green infrastructure for the reduction of coastal disasters: a review of the protective role of coastal forests against tsunami, storm surge, and wind waves. Coastal Engineering Journal63(3), 370-385.

[3] When reading the introduction, the authors refer interchangeably to tsunamis and storms. However, flooding and damage under both hazards present some differences and similarities. It would be interesting for readers to list them and explicitly mention which type of hazards will be the ones to be reproduced in the experimental design. Also, this needs to be translated into conclusions: are they valid for both tsunamis and storm surges or for one of them?

[4] Authors mention that the combination of submerged reefs/breakwaters with coastal trees has been studied previously. But the references given are very different from the design presented. They are usually combined as breakwater/reef in front of the forest and not as a base for each plant/tree. The authors should clearly mention if the proposed design is new or if it has already been described in the literature (reference 31 does not use reefs but a small concrete base for trees). In other words, background information on similar layouts or an explanation/justification of the proposed design is needed.

[5] Looking at the photos of the experimental setup, it looks like the simulated protection scheme consists of putting reef balls on each tree not only submerged but through the forest for each plant, even for emerged ones. Is this a realistic solution?

[6] The stated objective of the authors is (sic) to investigate experimentally the coastal flooding mitigation effect..... However, the authors do NOT show any data on coastal flooding. They simply compare how hydrodynamics are affected, but not how runup or inland extent is affected. Although related, other factors will condition/modulate flooding. Therefore, no direct conclusions on flooding can be drawn from the results obtained.

[7] One of the main parameters controlling the performance of a mangrove forest to protect inland against storm surge is not only the density but the width of the forest. Any comment on the dimensions of the forest in the experiment? How representative is? Also, how representative is the used density of the forest and how results can be extrapolated o other densities (if possible)? In the case that results cannot be generalized, authors need to state that their results are only valid for the tested range.

[8] Did the authors consider doing the experiment without trees (with reef balls only)? Depending on the length of the experiment runs, a large part of the observed changes in hydrodynamics could be associated with reefs. It would be interesting to know what the contribution of the reefs should be to the reduction as a sole protection measure.

[9] The authors mention that the physical scale of the experiment was 1:20. This is for the flow conditions (Froude). Did this also apply to the trees and reef balls or was a different scale used (what are the actual dimensions of the trees and reefs?) Any possible effect of scale on the interaction between flow and structures? [9] Can you indicate what is the magnitude of the modelled storm surge under prototype conditions?

[10] According to the results, flow height did not show a significant difference between studied cases. Which are the implications of this result in terms of flood risk?

[11] How much of the measured reflection is associated to the experimental layout? Will you expect a similar reflection coefficient in prototype conditions?

[12] The results obtained for reflection indicate that the greater the water depth (Hbore) at the reef ball locations, the lower the Cr. Since you are using emerged balls, this results in "artificially" low H* values. What will happen to submerged structures where H* will increase?

[13] The effect of initial ball water depth seems to be reproduced also in the results observed for RFI and RMI. As H* increases, these indices tend to approach for both designs (STDS is closer to RCDS).

[14] The authors claim that the tested RCDS have advantages over other composite defense systems, but do not provide any formal comparison against different criteria. Therefore, unless supported by formal analysis, this will be the sole opinion of the authors.

[15] The authors did NOT test or obtain any data related to the stability of each mangrove tree. Therefore, this cannot be a conclusion of this work.

[16] The results obtained are strictly valid for the range tested unless generalization is duly justified. Furthermore, the conditions tested are limited to reef balls located above mean water level and built around all the trees in the forest. This should be explicitly mentioned in the conclusions.

[17] Looking at the conclusions, it appears that the system tested has no limitations. The authors should clearly mention the limitations detected, the range of validity of the results obtained, and any other factor that could affect the validity of the scheme as a real solution (is it feasible to put a reef ball in each tree of a mangrove forest? what will happen if the density of the forest is different?). Just to illustrate real conditions, the trunks of mangrove trees are far from being tubular, they use to have a dendritic base in which it may not make much sense to surround it with a reef ball.

[18] Conclusions need to be fully reformulated and adapted just to obtained results.

Need extensive editing of English language and control of misprints/typos.

Author Response

Response to Reviewer’s comments:

We appreciate the valuable comments of the Reviewers on our manuscript. The following are the explanations presented in response to reviewers’ comments. The critical comments and useful suggestions have helped us to improve our paper considerably. As indicated in the responses that follow, we have taken all these comments and suggestions into account in the revised version of our manuscript and marked them with yellow color for easy tracking them.

 Response to Comments

Reviewer # 1: This is a manuscript presenting an experimental analysis of the performance of a hybrid (mangrove-reef balls) defense system again coastal flooding. The manuscript addresses a topic directly related to hydraulic (coastal) engineering which is one of the scientific targets of Water. In this sense, the manuscript can be of interest for Water readers. In what follow some comments/remarks are given.

Remark 1-1:  The manuscript needs an in-depth review of the grammar. This will help the authors to better transmit their results to readers. In addition to this, several (many) misprints (e.g. Tanka et al. instead of Tanaka et al.), uncompleted sentences (e.g. To compare the performance of the different compound defense systems two indices of ERFI and EMRI for drag force and moment reduction.), wrong selection of legends (Figure 1. Schematic sketch of experimental setup, (a) picture of designed quickly lifting gate. There is no indication of a and b in the figure). This needs to be fixed. So, please check carefully the text.  

Response 1-1:  The manuscript has been reviewed and all the suggested misprints, uncompleted sentences, and wrong selection of legends are corrected. Moreover, the manuscript is thoroughly proofread, its English is polished, and the corrections are highlighted with yellow color.

Remark 1-2:  There are some references on the topic that deserve to be included and, more importantly, considered to properly frame the presented work in the context of existing knowledge. Just as an example:

Chang, C. W., & Mori, N. (2021). Green infrastructure for the reduction of coastal disasters: a review of the protective role of coastal forests against tsunami, storm surge, and wind waves. Coastal Engineering Journal63(3), 370-385.

.Response 2:  Introduction section is revised and new references were added according to the comment. The revised part is highlighted with yellow color in page 2.

Remark 1-3:    When reading the introduction, the authors refer interchangeably to tsunamis and storms. However, flooding and damage under both hazards present some differences and similarities. It would be interesting for readers to list them and explicitly mention which type of hazards will be the ones to be reproduced in the experimental design. Also, this needs to be translated into conclusions: are they valid for both tsunamis and storm surges or for one of them?

Response 1-3:  The manuscript is revised accordingly: in Introduction (page 2,1st paragraph) the similarities and differences between tsunami and storm surge is mentioned.  The specified hazards is addressed both in Introduction (page 4, 3rd paragraph) and Materials and Methods (page 5,1st paragraph on). Then the validity of our proposed defence system for both tsunami and storm surge hazards are stated in Conclusions (page 22, 4th paragraph) and highlighted with yellow color.

Remark 1-4: Authors mention that the combination of submerged reefs/breakwaters with coastal trees has been studied previously. But the references given are very different from the design presented. They are usually combined as breakwater/reef in front of the forest and not as a base for each plant/tree. The authors should clearly mention if the proposed design is new or if it has already been described in the literature (reference 31 does not use reefs but a small concrete base for trees). In other words, background information on similar layouts or an explanation/justification of the proposed design is needed.

Response 1-4: The Introduction section is revised to reflect the previous studies on application of reef balls as submerged breakwaters with miniature reef balls in restoration programs for juvenile mangroves and highlighted with yellow color on page 4.

Considering novelty of the proposed design, to our best knowledge, this is the first study conducted to explore the dissipation of coastal flooding with the combining mangroves with reef balls. Therefore, it was emphasized in the Introduction section and highlighted with yellow color on page 4.

Remark1-5: Looking at the photos of the experimental setup, it looks like the simulated protection scheme consists of putting reef balls on each tree not only submerged but through the forest for each plant, even for emerged ones. Is this a realistic solution?

Response 1-5:   The main objective of this study is to evaluate experimentally how effective is the proposed Hybrid Defense System solution. Due to experimental limitation, the emerged forest type was conducted for two folds (i) to study the reduction of the velocity and height of incident surge and (ii) to investigate the absorbed drag force. The obtained results showed that the hybrid arrangement can increase the forest efficiency in reducing the incident flow velocity and promoting the absorbed drag force. However, as mentioned further studies are very necessary to examine the implementation of these structures in practical situations. As a realistic solution two aspects of feasibility and construction cost are important. The combination of reef balls and mangrove is practically feasible. For construction cost, a field study on creating structures in a strip with a limited width on the border of the mangrove forest should be estimated and then compared with the other alternatives. Therefore, aiming to attain this objective and considering the experimental limitations, present study was conducted on a low-density, emerged forest with a width of 0.9 meters, comparing the conditions of the forest, with and without the combination with reef ball modules.

Remark1-6: The stated objective of the authors is (sic) to investigate experimentally the coastal flooding mitigation effect..... However, the authors do NOT show any data on coastal flooding. They simply compare how hydrodynamics are affected, but not how runup or inland extent is affected. Although related, other factors will condition/modulate flooding. Therefore, no direct conclusions on flooding can be drawn from the results obtained.

Response 1-6:   To address the comment the title of the manuscript is changed to "Experimental Investigation on Coastal Flooding Hydrodynamics using a Hybrid Defense System" to better reflect the scope of this research work.

Remark1-7: One of the main parameters controlling the performance of a mangrove forest to protect inland against storm surge is not only the density but the width of the forest. Any comment on the dimensions of the forest in the experiment? How representative is? Also, how representative is the used density of the forest and how results can be extrapolated o other densities (if possible)? In the case that results cannot be generalized, authors need to state that their results are only valid for the tested range.

Response 1-7:   The manuscript is revised accordingly and specified this research limitation on pages 6 and 23 and highlighted with yellow color.

Remark1-8: Did the authors consider doing the experiment without trees (with reef balls only)? Depending on the length of the experiment runs, a large part of the observed changes in hydrodynamics could be associated with reefs. It would be interesting to know what the contribution of the reefs should be to the reduction as a sole protection measure.

Response 1-8: The study of reef balls only was out of concern of this work.   

Remark1-9:  The authors mention that the physical scale of the experiment was 1:20. This is for the flow conditions (Froude). Did this also apply to the trees and reef balls or was a different scale used (what are the actual dimensions of the trees and reefs?) Any possible effect of scale on the interaction between flow and structures? Can you indicate what is the magnitude of the modelled storm surge under prototype conditions?

Response 1-9:   As mentioned the physical scale was 1:20 and Froude's law of similarity was applied for the experiment. This scale was applied not only for flow conditions but also for trees and reef balls. The dimensions of the trees were based on the studies of Heidarzade [4] and Ghanavati [5] (tree height = 4.74 m, tree diameter = 0.186 m). Moreover, the dimensions of the reef ball modules were obtained from www.reefball.com (diameter = 1.83 m, height = 1.22 m). It is noted that the scaled dimensions for the models were rounded for convenience in their construction.

The potential effects of scaling on the flow-structure interaction in our study as the forest trees was not arranged with hyper-concentration can be negligible. However, for hyper concentration cases of the forest trees the scale reduction may bring the simulation closer to the real conditions. For example, in the study of Maza [6], a scale of 1:6 was implemented to reduce the scaling effects.

Regarding the magnitude of storm surge under prototype conditions, the estimated values were provided based on the dimensions of the land-falling wave, which is equal to    [8,9]. The simulated phenomenon in prototype conditions is expected to have a land-falling wave height ranges from 5.2 m to 12 m, inundation height ranging from 1.86 m to 3.45 m, and inundation velocity ranging from 3.55 to 13.17 m/s. Furthermore, comparing with the values estimated for surge induced by the Hurricane Katrina [10], the weakest simulated bore gives the land-falling wave height of 5 meters and inundation velocity of 5 m/s, that in turn result in a current with Froude number of 0.7.

Remark1-10: According to the results, flow height did not show a significant difference between studied cases. Which are the implications of this result in terms of flood risk?

Response 1-10:  The obtained results showed that there was not a significant difference in flow height between the cases which might be attributed to experimental setup and arrangement of reef balls and cylinder models.  Same observation is also reported by Zaha et al. [11] and Ahmed and Ghumman [13] for their hybrid defense system. In fact, in the process of energy transmission of flood surge other factors such as flow velocity and drag force absorption have a significant impact on the flood surge hydrodynamics. As mentioned in the manuscript, there was a remarkable decrease in flow velocity and increase in drag force absorption, which suggests that the flood energy reduces and the risk of flooding hazards may have decreased despite the negligible changes in the surge height. To clarify this point more, the manuscript is revised in the Conclusions accordingly and highlighted on pages 12 and 23.

Remark 1-11: How much of the measured reflection is associated to the experimental layout? Will you expect a similar reflection coefficient in prototype conditions?

Response 1-11:  A negligible reflection was observed in the case of STDS. This might be attributed to the structural complexity of actual mangrove forests and its hyper-concentrated flow behaviour. As noted in the previous studies, the physical structure of vegetation is a crucial factor in determining the efficacy of coastal forests on wave attenuation [6]. Therefore, our simplified model may face shortcoming in fully capture the behaviour of mangrove forests. Conversely, a reflected wave was observed very clearly in the RCDS case. In RCDS case the frontal area ( ) of the model was increased significantly, which caused a portion of the flow energy is reflected in the form of a surge towards the upstream direction. In prototype conditions, however, an increase in  due to the presence of actual mangroves is likely lead to higher reflected wave heights and subsequently higher reflection coefficients in both STDS and RCDS cases. The reflection coefficients depend on the forest density and structural complexity of hyper-concentrated flow. Further research is needed to provide better insight into the behaviour of mangrove forests under real-world conditions.

Remark 1-12: The results obtained for reflection indicate that the greater the water depth (Hbore) at the reef ball locations, the lower the Cr. Since you are using emerged balls, this results in "artificially" low H* values. What will happen to submerged structures where H* will increase?

Response 1-12:  As mentioned, the increase in water depth ( ) at the reef ball locations leads to higher H* values in the submerged conditions. Consequently, the flow interaction with the reef ball structures decreases and lower values for the reflected wave height is expected, and the reflection coefficient ( ) decreases. However, the change in H* values due to the degree of submergence is a critical factor and needs to be explored in further investigations.

Remark 1-13: The effect of initial ball water depth seems to be reproduced also in the results observed for RFI and RMI. As H* increases, these indices tend to approach for both designs (STDS is closer to RCDS).

Response 1-13:  From another angle the hydrodynamics of hybrid defense system can be described by the momentum reduction and drag force absorption. In the case of STDS, as the H* increases, more frontal area ( ) of the trees interacts with the incident surge, resulting in an increase in the amount of drag force absorption and momentum reduction. This increase leads to higher maximum reduction rates of the drag force index (RFI) and maximum reduction rates of the moment index (RMI). On the other hand, the main performance of the RCDS in drag force and momentum absorption resulted by the presence of reef ball structures. As the reef ball modules are submerged in all  cases, by increasing of H*,  of RCDS slightly increases. Hence, as the H* increases, the improvement of performance of the RCDS in absorbing drag force and moment reduction is relatively small. It is expected that the RFI and RMI values remain almost constant with increasing H* in the RCDS design.

Remark 1-14: The authors claim that the tested RCDS have advantages over other composite defense systems, but do not provide any formal comparison against different criteria. Therefore, unless supported by formal analysis, this will be the sole opinion of the authors.

Response 1-14: Table 1-1 shows the comparison of momentum reduction and drag force absorption of the presented RCDS in our study and the CDSs proposed by Zaha et al. [11] and Kimiwada et al. [12]  As seen, the evaluation indices (ERFI and ERMI) for CDSs have higher values in comparison with the STDS, and the RCDS model showed 60% and 65% more effectiveness for ERFI and ERMI compared to the STDS model, respectively. Comparing with the previous combination methods presented by Zaha et al. [11] and Kimiwada et al. [12], the presented RCDS model has an acceptable performance in the same order as the compound systems offered by Zaha et al. [11] and produces a better performance than the proposed combination of Kimiwada et al. [12]. Meanwhile, having the lower construction cost (no need to vacant land for construction) and potential of increasing the stability of trees make RCDS a preferable compound defense system over a combination of vegetation with embankment and moat.”

Table 1-1: Comparison of ERFI and ERMI

Case

ERFI

ERMI

STDS

51.2

49.8

RCDS

81.8

82.55

V40ME [11]

74.3

92.4

EMV40 [11]

83.1

90.2

EMVS40 [12]

68.7

80.8

EVS40M [12]

66.7

74.6

The manuscript is revised accordingly on pages 20 and 22 and highlighted with yellow colour.

Remark 1-15: The authors did NOT test or obtain any data related to the stability of each mangrove tree. Therefore, this cannot be a conclusion of this work.

Response 1-15: The manuscript is revised accordingly in the Conclusion section and highlighted with yellow colour.

Remark 1-16: The results obtained are strictly valid for the range tested unless generalization is duly justified. Furthermore, the conditions tested are limited to reef balls located above mean water level and built around all the trees in the forest. This should be explicitly mentioned in the conclusions.

Response 1-16: The manuscript is revised accordingly in the Conclusion section and highlighted with yellow colour on page 23.

Remark 1-17: Looking at the conclusions, it appears that the system tested has no limitations. The authors should clearly mention the limitations detected, the range of validity of the results obtained, and any other factor that could affect the validity of the scheme as a real solution (is it feasible to put a reef ball in each tree of a mangrove forest? what will happen if the density of the forest is different?). Just to illustrate real conditions, the trunks of mangrove trees are far from being tubular, they use to have a dendritic base in which it may not make much sense to surround it with a reef ball.

Response1-17: The hybrid defense system as a combination of mangrove trees with reef ball modules was introduced and experimentally and the obtained results are applicable to the tested range. On the other hand, the trunks of mangrove trees are far from being tubular and have a dendritic base. It is expected that the mangrove roots will emerge out of the reef ball holes on its after plantation and during growth and formation. Thus, the structure of the tree and the reef ball will act in combination with each other to enhance the overall resilience of the mangrove ecosystem.

The manuscript is revised accordingly in the Conclusion section and highlighted with yellow colour.

Remark 1-18: Conclusions need to be fully reformulated and adapted just to obtained results.

Responcse1-18: The conclusion has been totally revised and formulated according to the Reviewer’s comments and highlighted with yellow colour.

Reviewer 2 Report

The manuscript is dealing with the interesting topic of green infrastructures for coastal protection from hydraulic disasters. The experimental methods are clearly explained but the original data should be also demonstrated for the calculation of bore speed, which may be calculated the time series of ultrasonic sensor data. If such a data is included in the supplementary material, but there are no files in the page that was mentioned in the paper.

Author Response

Response to Reviewer’s comments:

We appreciate the valuable comments of the Reviewer on our manuscript. The following are the explanations presented in response to reviewers’ comments. The critical comments and useful suggestions have helped us to improve our paper considerably. As indicated in the responses that follow, we have taken all these comments and suggestions into account in the revised version of our manuscript and marked them with yellow color for easy tracking them.

 Response to Comments

Reviewer # 2: The experimental methods are clearly explained but the original data should be also demonstrated for the calculation of bore speed, which may be calculated the time series of ultrasonic sensor data. If such a data is included in the supplementary material, but there are no files in the page that was mentioned in the paper.

Response: 

The time series of ultrasonic sensor data for the case of  "H_t = 0.5 m"  and "H_c = 0.14m" is included in the supplementary material for the submission of the revised manuscript.

Round 2

Reviewer 1 Report

Discussion

The authors should comment on how to implement this type of measurement in real conditions where the logs do not have a simple and homogeneous tubular shape. Mangroves often have dendritic roots that deviate significantly from the idealized shape tested by the authors. This may have implications for reef mating. Also, from a constructive point of view, how should they be built?

Conclusions

The last conclusion is not consistent with that of the second point, where no significant differences in flow depth were found. Reading the last sentence it seems that the tested measure is a good solution for urbanized areas. However, to support this conclusion, the authors should test a much wider range of forcing conditions and test a more realistic design. Without including an analysis of the width required to reduce the hazard/impact/ damage for a given wave/tsunami of given magnitude, this conclusion cannot be objectively supported.

Author Response

Remark 1-1:  The authors should comment on how to implement this type of measurement in real conditions where the logs do not have a simple and homogeneous tubular shape. Mangroves often have dendritic roots that deviate significantly from the idealized shape tested by the authors. This may have implications for reef mating. Also, from a constructive point of view, how should they be built?

Response 1-1:  The complex shape of mangrove root presents many construction challenges. The potential solution to tackling the challenges is to plant mangroves in reef balls or place them around existing young/small mangroves on the coastal site. This approach provides a flexible means of accommodating the natural shapes of the legs and roots, while promoting mangroves expansion and resilience. The proposed design contributes to reducing the coastal hazards; however, further research is very necessary to look for both an excellent accommodation method to enhance the mangrove resilience and expansion ranges of forest width.  Proper planning and execution are also critical for ensuring that mangroves and reef balls are integrated effectively into coastal management strategies to mitigate flood risk.

The manuscript is revised accordingly on page 22 and highlighted with yellow color.

Remark 1-2:  The last conclusion is not consistent with that of the second point, where no significant differences in flow depth were found. Reading the last sentence it seems that the tested measure is a good solution for urbanized areas. However, to support this conclusion, the authors should test a much wider range of forcing conditions and test a more realistic design. Without including an analysis of the width required to reduce the hazard/impact/ damage for a given wave/tsunami of given magnitude, this conclusion cannot be objectively supported.

.Response 1-2:  The Conclusion section is revised accordingly and highlighted with yellow color.

Reviewer 2 Report

The manuscript has been well revised with the attachement data. The additional sentences in the introduction, material methods and conclusions improved the manuscript with much clear explanations of the target of this experiments. It seems no problemse for publication whole through the docments.

Author Response

Remark 2-1:  The manuscript has been well revised with the attachment data. The additional sentences in the introduction, material methods and conclusions improved the manuscript with much clear explanations of the target of this experiments. It seems no problems for publication whole through the documents.

Response 2-1:  We are most grateful for the valuable comments on our manuscript; the useful suggestions have helped us to improve our manuscript considerably.